# Incidence of Bacteremia Consequent to Different Endoscopic Procedures in Dogs: A Preliminary Study

**DOI:** 10.3390/ani10122265

**Published:** 2020-12-01

**Authors:** Alba Gaspardo, Maria Chiara Sabetti, Renato Giulio Zanoni, Benedetto Morandi, Giorgia Galiazzo, Domenico Mion, Marco Pietra

**Affiliations:** Department of Veterinary Medical Sciences, Alma Mater Studiorum–University of Bologna, via Tolara di Sopra 50, 40064 Ozzano Emilia, BO, Italy; alba.gaspardo2@unibo.it (A.G.); mariachiara.sabetti2@unibo.it (M.C.S.); renatogiulio.zanoni@unibo.it (R.G.Z.); giorgia.galiazzo2@unibo.it (G.G.); domenico.mion2@unibo.it (D.M.); marco.pietra@unibo.it (M.P.)

**Keywords:** endoscopy, bacteremia, incidence risk, risk ratio, dogs, veterinary teaching hospital

## Abstract

**Simple Summary:**

Antimicrobial resistance is a threat that poses a great risk to public health. It has been predicted that, by 2050, there will have been 10 million deaths worldwide due to drug-resistant infections. There is a crucial need in Veterinary Medicine to reduce the use of antimicrobials to slow down the process and incidence of antimicrobial resistance as a One Health concern. The aim of this study was to evaluate the appearance of bacteremia following endoscopic procedures in dogs brought to the Veterinary Teaching Hospital (VTH) of the Department of Veterinary Medical Science of the University of Bologna. The results obtained from hemocultures before and after the endoscopic procedures demonstrated a low incidence of bacteremia after endoscopy. This could be seen as an attempt to reduce the use of antimicrobials to avoid the spread of antimicrobial resistance.

**Abstract:**

Endoscopic procedures are widely used in veterinary medicine, and their role in producing transient bacteremia is debatable. The growing issue of antibiotic resistance requires the correct use of antibiotics, avoiding their administration when not strictly necessary. Studies highlighting post-endoscopy bacteremia in veterinary medicine are extremely rare and often involve very few animals. This study describes the results from 74 owned dogs, brought to the Veterinary Teaching Hospital of the Department of Veterinary Medical Science of the University of Bologna, for the purpose of undergoing an endoscopic procedure. Two blood samples were taken from each dog, one before and one after the procedure, in order to assess the incidence of bacteremia linked to endoscopic procedures. Eight dogs were tested positive at the second blood culture with an Incidence Risk (IR) of 10.8%. No statistical differences were found by comparing positive and negative blood cultures with respect to sex, age, weight and anesthesia duration. In addition, no difference was found between airway and digestive tract procedures. The present findings showed that the probability of developing bacteremia after an endoscopic procedure was quite low, and additional studies confirming this are certainly recommended as well as the evaluation of categories of patients potentially considered at risk.

## 1. Introduction

Endoscopic procedures in veterinary medicine are performed for both diagnostic (i.e., gastrointestinal and respiratory disease) and therapeutic (such as the removal of foreign bodies) purposes [1,2,3,4,5,6].

Local or remote tissue infections are complications that may follow an endoscopy. These could be the result of mucosal trauma with consequent alteration of the physiological barriers, and bacterial translocation of endogenous microbial flora into the bloodstream [7,8].

The growing issue of antimicrobial resistance strengthens the necessity of correctly using these drugs, avoiding their administration when not strictly needed [9,10]. The incidence of bacteremia linked to endoscopic procedures has been assessed in human medicine by several studies which showed a low percentage, up to 8%, of short-lasting transient bacteremia in these patients [7,11,12,13]. Considering these results, it is possible to draw up guidelines, in human medicine, for antibiotic prophylaxis during endoscopies that permit administration of antibiotics only in patients considered at risk (patients with high-risk cardiac conditions or gastrointestinal tract infections) or undergoing procedures considered at high risk (esophageal bougienage or PEG/PEJ tube placement) [8].

Studies regarding post-endoscopy bacteremia in veterinary medicine are extremely rare [14,15]. Therefore, there are no specific guidelines regarding preventive antibiotic prophylaxis for dogs and cats undergoing endoscopic procedures, leaving the decision up to the clinician.

The aim of this preliminary prospective study was to evaluate the incidence of bacteremia related to endoscopies in dogs, to assess whether or not antibiotic prophylaxis is required for such procedures in veterinary medicine.

## 2. Materials and Methods

Owned dogs undergoing endoscopic procedures, brought to the Veterinary Teaching Hospital (VTH) of the University of Bologna from November 2015 to June 2017, were included in the study. The endoscopic procedures considered were either for diagnostic or therapeutic purposes: rhinoscopy, bronchoscopy, gastroscopy and gastro-duodeno-colonoscopy.

Dogs over six months of age and weighing more than 3 kg (kg) were included. All dogs underwent routine pre-anesthetic hematology tests, carried out using an automated hematology system (ADVIA 2120, Siemens Healthcare Diagnostics, Tarrytown, NY, USA). Dogs having anemia, immunodeficiency or sepsis based on the reference intervals provided by Moritz et al. [16] as well as those having concurrent pathologies or symptoms except those related to the endoscopy were excluded. Dogs who had undergone immunosuppressive and/or antibiotic treatments within the previous three weeks were also excluded, except for those in therapy with tylosin.

To assess the incidence of bacteremia, as the result of mucosal trauma, linked to endoscopic procedures, two blood samples were taken for blood culture, the first immediately before the beginning of the procedure and the second twenty minutes after the end of the procedure. Both blood samples were taken with the venous catheter used for the anesthesia in order to eliminate additional painful and stressful manual manipulation.

To allow adequate functioning of the blood culture system chosen for the study (Signal™ Blood Culture System (Oxoid) Waltham, MA, USA), the minimum quantity of blood to be taken was 3 mL and the maximum was 8 mL. Blood volumes were established based on the welfare of the dogs and the blood culture system limit threshold.

The circulating blood volume in a dog is approximately 85 mL per kilogram (mL/kg) of body weight. The maximum amount of blood that can be collected from a dog for a transfusion donation, without causing hemodynamic alterations, is 19 mL/kg corresponding to approximately 17–22% of the total blood volume [17]. Therefore, in order not to cause any suffering, dogs weighing less than 3 kg were excluded from the study and, for those between 3 kg and 10.5 kg, the minimum amount of blood required for the blood culture system was collected at each time point. The dogs were divided into three groups based on body weight: (1) 3–10.5 kg, (2) 10.6–25 kg, (3) >25 kg. For each group, a different amount of blood was drawn: 3 mL in Group 1; 5 mL in Group 2; 8 mL in Group 3.

Each step was performed under sterile conditions by a trained operator, and the blood samples were obtained by means of a venous catheter allocated for the general anesthesia. The anesthetic protocol for each dog was chosen based on the temperament of the animal itself and the type of procedure required.

To position the intravenous catheter, the hair above the cephalic vein was removed. Subsequently, the skin was disinfected with three alternating passages of 4% chlorhexidine and denatured ethyl alcohol 70% (each left on for 30 s), following the main indications for skin antisepsis [18,19]. Once the area was ready, the intravenous catheter was positioned with the operator wearing sterile gloves. Following the manufacturer’s instructions, the blood samples were collected, and were immediately aseptically injected into the Signal™ Blood Culture System (Oxoid) and incubated at 36 ± 1 °C.

The blood cultures were routinely inspected twice a day for a total incubation period of at least 7 days. A positive result was indicated when the blood/broth mixture was forced into the signal device by the pressure created by growing organisms. The positive blood cultures were subcultured by streaking 20 µL of the blood/broth mixture onto one plate of Blood Agar (BBL) and were incubated in aerobiosis. Furthermore, two plates of Columbia Blood Agar (BD) were incubated in anerobiosis and in microerophilia, respectively. The incubation temperature was 36 ± 1 °C for all plates. To isolate any metabolically weak bacteria which were not able to activate the signal device, blind subcultures were obtained from the negative blood cultures at the end of the 7th day by applying the same conditions used for the positive ones. All the bacterial strains isolated were identified first phenotypically using Analytic Profile Index (API) test systems (bioMérieux) and were then confirmed genotypically by partial sequencing of the 16S rRNA gene [20].

The trial was authorized by the Animal Welfare Committee of the University of Bologna (Protocol N. 1173 of 11/06/2020).

### Statistical Analysis

The data collected at the time of hospitalization were entered into a, MS Excel spreadsheet, merged with the microbiological results, then imported into Stata 15 (StataCorp LLC, College Station, Texas, TX, USA) for the analyses. Sex, age by month, weight, anesthesia duration and endoscopic procedures were the variables grouped. The endoscopies were first analyzed as a dichotomous variable, being divided into the airway and digestive tract procedures. Thus, 5 groups were formed and assessed: gastroscopy, gastro-duodeno-colonoscopy, rhinoscopy, bronchoscopy and rhinoscopy + bronchoscopy. Incidence risk was interpreted as the probability of a dog developing bacteremia owing to the endoscopic procedure. The non-normally distributed continuous data were summarized using medians and interquartile ranges (IQRs), while the normally distributed data were summarized using mean ± SD. When reasonable, the continuous variables were categorized. Pearson’s χ2 test was used to compare the hemoculture results and the categorical variables. The Fisher’s exact *P*-value was considered as more than 20% of the cells had expected frequencies <5 [21]. The Risk Ratio (RR) and relative 95% confidence interval (CI) were also calculated to evaluate whether tylosin was a protective factor for developing bacteremia. The Kruskal–Wallis non-parametric test was carried out when non-normal distributed continuous variables were involved. The results were considered to be significant when *p* ≤ 0.05.

## 3. Results

Overall, seventy-four dogs were included in the study. A total of 148 hemocultures were carried out. There were 31 males (42%), 17 females (23%), 7 castrated males (9%) and 19 neutered females (26%) included, the median age was 46 (IQR 21–86) months (range 5–179). The weight ranged from 3.7 to 56 kg (mean 22.2, SD + 10.1). Twelve dogs belonged to Group 1 (16%), 33 to Group 2 (45%) and 29 to Group 3 (39%). Ten out of the 74 dogs were under tylosin treatment (13.5%). Blood samples were obtained before and after performing 74 endoscopic procedures which included 20 gastroscopies (27%), 17 gastrointestinal endoscopies (23%), 18 rhinoscopies (24.3%), 11 bronchoscopies (14.9%) and 8 rhinoscopies + bronchoscopies (10.8%). The median duration of the anesthesia procedures was 45 min (min 15–max 240, IQR 30–80). The distribution of the different variables is summarized in Figure 1.

Only dogs with a negative pre-procedure blood culture and a positive post-procedure blood culture were considered as positive for the statistical analysis.

Patients in which pre-procedure results were positive and the post-procedure results were negative or positive were considered to be negative for the statistical analysis, since the results were not attributable to the endoscopic procedure (see Table 1).

A total of 66 dogs were negative. Of those, twenty-seven were males (41%), 17 females (26%), 16 neutered females (24%) and 6 castrated males (9%). The median age was 44 (IQR 21–90) months (range 5–179). The mean weight was 22.6 ± 10.1 kg (3.7–56). Finally, the median length of anesthesia was 47.5 (IQR 30–80) mins (range 15–240).

Eight dogs were positive at the second blood culture (10.8%). Four males, one castrated male and three neutered females were bacteremia-positive. The mean weight was 12.2 ± 10.6 kg (range 8.8–40.7), the median anesthesia duration was 34.5 (IQR 22.5–80), ranging from 20 to 100 min. The median age was 60.5 (IQR 33.5–82.5) months, ranging from 21 to 149 months. The bacteria isolated were: *Pseudomonas aeruginosa* (3 dogs), *Propionibacterium acnes* (2 dogs), *Staphylococcus warneri* (1 dog), *Staphylococcus pseudointermedius* (1 dog), *Aeromonas hydrophila* (1 dog).

When comparing the positive and the negative blood cultures regarding sex, age and weight, no statistical differences were found. Endoscopic procedures were analyzed both as dichotomous and as categorical and none of the variables showed any statistical difference. All variables, values and results are summarized in Table 2.

Ten dogs were under treatment with tylosin at the time of the procedure; relative risk was calculated as RR = 0.914, resulting in a weak protective factor but still not significant (95% CI: 0.13–6.66).

Finally, based on the non-parametric Kruskal–Wallis test (*h* = 0.933, *p* = 0.333), the anesthesia duration did not statistically affect the probability of developing bacteremia after the procedure.

## 4. Discussion

The aim of this prospective study was to evaluate the incidence risk of bacteremia following endoscopic procedures of the respiratory and gastrointestinal tracts.

The first sampling was useful in demonstrating that the dog was not previously bacteremic before endoscopy. In fact, transient bacteremia can arise following ordinary daily activities without causing any symptoms [22].

Eight subjects showed negative blood cultures before the procedure and were positive at the end of the endoscopy. These cases were considered as bacteremic consequent to the execution of the endoscopic procedure. The bacteria isolated were found to be normal inhabitants of cutaneous and upper airway microbial flora. Thus, it was reasonable to hypothesize that these positive results could be associated with mucosal trauma and subsequent translocation of native microorganisms. There were no significant differences concerning the onset of bacteremia regarding sex, age, body weight, procedure duration or type of endoscopy. Furthermore, eight patients were positive for the same bacterial strain in both blood cultures, and five patients were positive at the first blood culture and negative at the second. These patients were considered negative as the positivity was not considered to be related to the endoscopic procedures. In these dogs, the isolated microorganisms were found to be bacteria belonging to normal canine microflora or contaminants, such as *Serratia marcescens* [23]. Even using aseptic manual manipulation, a certain percentage of blood culture contamination is also described in human medicine [24,25,26].

Dogs who had taken antibiotics or immunosuppressants in the 20 days prior to the endoscopy were excluded from the study, with the only exception being tylosin therapy. Until recently, tylosin was part of the therapeutic and diagnostic trial of canine chronic enteropathies [27]. After the tylosin trial, non-responsive patients underwent gastro-duodeno-colonscopy without suspending the tylosin-treatment. In order to not exclude this wide group of patients, we felt to consider them for analysis. However, the results reported that tylosin had a weak protective factor (RR = 0.914) for developing bacteremia after the procedure, but it was still not significant.

Antimicrobial prophylactic treatment, in human medicine, is only recommended for patients considered at risk of developing complications following transitory bacteremia, such as infectious endocarditis. The high-risk category includes patients suffering from severe cardiac conditions, those who are immunocompromised or those undergoing high risk-procedures, such as esophageal dilation and sclerotherapy [28,29]. In human medicine, performing gastrointestinal or respiratory tract endoscopies did not statistically affect the incidence risk of developing bacteremia [11,12,13].

The authors’ findings showed that the probability of developing bacteremia consequent to endoscopic procedures of the respiratory and gastrointestinal tracts in the dog population in this study was quite low, with an incidence risk of 10.8%. This is in agreement with the American Society for Gastrointestinal Endoscopy which reports gastroscopy in humans as a low risk-procedure with an incidence of bacteremia ranging from 0% to 8%, with or without carrying out a biopsy [28]. To the authors’ knowledge, only two similar studies have been carried out on dogs, reporting an incidence of 17.9% and 3%, respectively [14,15]. Li et al. [14] used dogs as an animal model to assess bacteremia after Endoscopic Ultrasonography-guided Fine Needle Aspiration (EUS FNA) of the pancreas in humans. Authors reported an incidence of bacteremia around 18%. The higher value compared to our one was perhaps due to the more invasive procedure performed. Although the dogs utilized were experimental-dogs, a high frequency of contamination was observed as a consequence of the high amount of bacteria present on the dog’s hair which can translocate through the intravenous catheter. The second study [15] aimed to evaluate whether the use of proton pump inhibitors in dogs favored the onset of bacteremia following a gastroscopy. Differences in the incidence of bacteremia could have been due to the low number of samples included in the other study and to the inclusion of only gastroduodenoscopies as compared to the variable procedures in the present study. Moreover, Jones et al. [14] used research dogs having almost the same body weight, and the same amount of blood was drawn for each culture. Finally, the samples were taken without the aid of an intravenous catheter.

After performing the endoscopy, the choice of whether to start antibiotic therapy was evaluated by the veterinarian, based on the animal’s endoscopic diagnosis and clinical situation, and not on the result of blood culture. None of the patients developed symptoms attributable to infections or septicemia after the procedures.

The study had some limitations. First, an additional blood culture was not carried out hours after the end of the endoscopy to evaluate the short duration of the bacteremia caused by the procedures. In the majority of cases, bacteremia caused by endoscopic procedures is of short duration and is most frequently observed within the first 30 min after the end of the procedure [7,22]. Since the animals in this study were not experimental animals, after the complete awakening from the anesthesia, the intravenous catheter was removed, and the patients were discharged from the hospital. Therefore, a third blood culture, could not be carried out using the venous catheter; moreover, another blood sample would have increased the amount of blood taken from each animal, impacting the dog’s hemodynamics. To eliminate additional painful and stressful manual manipulation, it was decided to use the intravenous catheter in order to minimize the impact on the animals, even if this represented another limitation. In fact, transient bacteremia can occur due to various causes, including the insertion of an intravenous catheter. Despite maintaining sterile conditions, the intravenous catheter could also be a bacterial adhesion site [30,31]. The choice of reducing the amount of blood for each blood culture, related to the well-being of the dogs, was an additional limitation as it may have affected the result of the blood cultures. The quantity of blood and the timing and methods of incubation are, in fact, important for obtaining a reliable result when carrying out a blood culture [32]. However, to minimize the occurrence of false-negative results, we used a strong conservative approach re-culturing negative blood cultures under the same conditions as the positive ones.

## 5. Conclusions

In conclusion, to the best of the authors’ knowledge, there are no guidelines regarding preventive antibiotic therapy following these endoscopic procedures in dogs. These preliminary results tended to exclude the preventive use of antibiotics following endoscopic procedures, unless related to the disease diagnosed. However, additional studies confirming the low incidence and transient nature of the bacteremia are certainly recommended, as is including and evaluating the categories of dogs potentially considered at risk.

## Figures and Tables

**Figure 1 animals-10-02265-f001:**
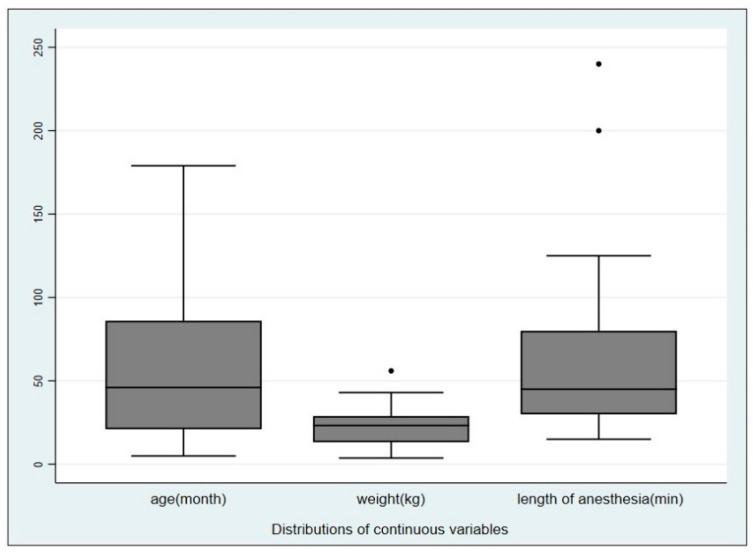
Boxplot of age, weight and length of the anesthesia procedures in the population sampled, displaying the distribution of data.

**Table 1 animals-10-02265-t001:** Interpretation of the hemoculture results attributable to the procedures.

Animals	Pre-Procedure Result	Post-Procedure Result	Procedure Attributable Result
n = 53	−	−	−
n = 8	+	+	−
n = 5	+	−	−
n = 8	−	+	+

+ tested positive to the blood cultures; − tested negative to the blood cultures.

**Table 2 animals-10-02265-t002:** Pearson χ^2^ comparison between the categorical variables and the hemoculture results.

Categorical Variables	Categories	# of Patients	%	Pearson χ2	Fischer’s Exact *p*-Value	Degrees of Freedom
Sex	Male	38	51.3	0.446	0.387	1
	Female	36	48.7			
Age	≤12 months	9	12.2	4.2351	0.273 *	3
	>12 ≤57	34	45.9			
	>57 ≤84	12	16.2			
	>84	19	25.7			
Weight	≥3 kg ≤10.5 kg,	12	16.0	0.946	0.614 *	2
	≥10.6 kg <25 kg,	33	45.0			
	≥25 kg	29	39.0			
Endoscopy procedures (dichotomous)	airways	37	50.0	2.242	0.261	1
	digestive tract	37	50.0			
Endoscopy procedures (categorical)	gastroscopy	20	27.0	3.724	0.284 *	4
	gastro-duodeno-colonoscopy	17	23.0			
	rhinoscopy	18	24.3			
	bronchoscopy	11	14.9			
	rhino+broncho	8	10.8			

* Overall *p*-value.

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
