# Peer review of "Incidence of Bacteremia Consequent to Different Endoscopic Procedures in Dogs: A Preliminary Study"

_animals, 2020, doi:10.3390/ani10122265_

Round 1

Reviewer 1 Report

The publication is interesting and deals with a very current topic: the correct use of antibiotics to reduce antibiotic resistance. the text is well structured and every aspect is treated clearly. I only found the following inconsistencies:
1) line 76-77 reads 3ml in group 1; 5ml in group 2; 8 ml in group 3 but in lines 176-177 we read the minimum quantity of blood to be taken was 3ml and the maximum was 10ml - clarify whether the maximum quantity was 8 or 10ml;
2) lines 190-192 we read that 8 patients were positive at both the first and second sampling - but this is not stated in the results - and that 5 patients were positive at the first sampling and negative at the second but in the results and in Table 2 patients positive at the first sampling and negative at the second are 8 - clarify

Author Response

The publication is interesting and deals with a very current topic: the correct use of antibiotics to reduce antibiotic resistance. the text is well structured and every aspect is treated clearly. I only found the following inconsistencies:

Thank you for the real positive comment

1) line 76-77 reads 3ml in group 1; 5ml in group 2; 8 ml in group 3 but in lines 176-177 we read the minimum quantity of blood to be taken was 3ml and the maximum was 10ml - clarify whether the maximum quantity was 8 or 10ml;

Agreed, we were wrong making confusion with the maximum amount that Signal™ Blood Culture System (Oxoid) can hold. We have changed the sentence Lines 176-177 as follow: “taken was 3 ml and the maximum was 8ml”

2) lines 190-192 we read that 8 patients were positive at both the first and second sampling - but this is not stated in the results - and that 5 patients were positive at the first sampling and negative at the second but in the results and in Table 2 patients positive at the first sampling and negative at the second are 8 – clarify

Agreed, we have added into the results section that positive patients at both cultures were considered as negative, and also have added a row into Table 1 reporting the number of positive patients pre and post-procedure.

Reviewer 2 Report

Overview 

The manuscript “Incidence of bacteremia consequent to different endoscopic procedures in dogs: a preliminary study” by Gaspardo et al., intended to report the incidence of bacteremia following endoscopic procedures. Although it shows novelty, since the other published study regarding the same issue only included 8 animals, I have some concerns regarding the methodology used, such as the method for colleting the blood samples which could increase the samples contamination; the time of blood collection after the endoscopy which was only one immediately after the procedure; the fact that it was included animals doing an antibiotic (tylosin) which can bias the results; and also the culture procedures which were not specified for what type of bacteria – aerobic and/or anaerobic.

Specific comments

Simple summary        

Lines 20-21: I believe that the statement “this is a first step towards the reduced use of 
antimicrobials” is exaggerated because so far there no indications to use antibiotics following endoscopy in veterinary medicine, unless there is any clinical signs that justifies it, and at most this study concludes that the incidence of bacteremia after endoscopy is low but it does not evaluates if the bacteremia is transient and does not reports the follow-up of these dogs to know if they actually developed any clinical symptoms related to it or not. So I suggest that it should be rephrased or deleted.

Abstract

Line 24: There is only one veterinary study according to this manuscript introduction, so it’s not correct to mention “studies” and “often”. Please be more accurate in your descriptions.

Introduction

Lines 39-40: Please correct “i.e. gastrointestinal 
or respiratory disease” by for example gastrointestinal 
and respiratory disease, because there are many other endoscopic procedures.

Line 47: Please specify what is how low is the percentage of transient bacteremia in humans after endoscopy.

Lines 49-50: Please specify which patients are considered at risk and what procedures for antibiotic prophylaxis.

Line 51: Again if the author just mentioned one reference why this sentence says “Studies…” And what showed this particular study?

Line 55: Please delete “to add data”

Lines 55-56: The authors mention as a goal of this study: “assess whether or not antibiotic prophylaxis is required 
for such procedures in veterinary medicine” – but in fact the authors did not study that. They did not follow these patients clinically not even verified if the bacteremia was transient or persistent.

M&M

Line 63: The author’s mention that Dogs over six months…were potentially included. What does this mean exactly? Were they included or not?

Line 69: The authors mention as a exclusion criteria the administration of antibiotics within the previous three weeks but then say that included dogs doing tylosin… This does not makes sense at all. These dogs should be obviously excluded from the study. This is an important bias for the results.

Lines 70-72: Why twenty minutes and not 30 min, 60, 120 minutes and/or 24h hours after? Is 20 minutes sufficient to bacteria translocate in enough numbers to be cultured? Is there any scientific data to support your choice? How do the authors confirm if the bacteremia is transient or persistent? And therefore, how can the authors judge about the need (or not) of prophylactic antibiotics??

Lines 88-89: What type of cultures: aerobic, anaerobic or both?? Please clarify.

Results

Line 135: The authors mention that there were 8 patients in whom preprocedure results were positive and the postprocedure results were negative. How do the authors explain these tests results? Could they be false positives or false negatives? It seems to me that these results seriously question the culture methodology…

Line 157: Table 2: please insert the data regarding the number and percentage of cases for each variable

Discussion

Line 191-195: Taking in consideration the high number of these cases (the same number of positive cases) with contaminants as the author said, we may think that the collecting technic perhaps was not the most appropriate… and this may invalidate the results of this study. Any other explanation?

Lines 220-221: Comparing this study with others, the authors mention that samples were taken from the intravenous catheter. Considering the results of the present study, it seems more feasible to avoid collecting the blood from a catheter used for other purposes, such as inject anesthesia medications. To me this increases the risk of getting a contaminated sample which again bias the results of this study.

Author Response

Overview 

The manuscript “Incidence of bacteremia consequent to different endoscopic procedures in dogs: a preliminary study” by Gaspardo et al., intended to report the incidence of bacteremia following endoscopic procedures. Although it shows novelty, since the other published study regarding the same issue only included 8 animals, I have some concerns regarding the methodology used, such as the method for colleting the blood samples which could increase the samples contamination; the time of blood collection after the endoscopy which was only one immediately after the procedure; the fact that it was included animals doing an antibiotic (tylosin) which can bias the results; and also the culture procedures which were not specified for what type of bacteria – aerobic and/or anaerobic.

Thank you for having had time to review the manuscript. Hopefully, we have clarified and detailed point-by-point your concerns.

Specific comments

Simple summary        

Lines 20-21: I believe that the statement “this is a first step towards the reduced use of 
antimicrobials” is exaggerated because so far there no indications to use antibiotics following endoscopy in veterinary medicine, unless there is any clinical signs that justifies it, and at most this study concludes that the incidence of bacteremia after endoscopy is low but it does not evaluates if the bacteremia is transient and does not reports the follow-up of these dogs to know if they actually developed any clinical symptoms related to it or not. So I suggest that it should be rephrased or deleted.

The sentence has been rephrased: “this could be seen as an attempt to reduce the use of antimicrobials to avoid the spread of antimicrobial resistance”

Abstract

Line 24: There is only one veterinary study according to this manuscript introduction, so it’s not correct to mention “studies” and “often”. Please be more accurate in your descriptions.

We have added an other reference which evaluates bacteremia in dogs after endoscopic ultrasonography-guided: Li, N.; Qi, H.; Liu, Z.; Ge, N.; Guo, J.; Wang, G.; Liu, X.; Wang, S.; Sun, S. Effect of povidone-iodine washing of gastrointestinal mucosa or taking proton pump inhibitors on bacteremia after endoscopic ultrasonography-guided fine needle aspiration. Endosc Ultrasound 2012 1, 90-95.

Introduction

Lines 39-40: Please correct “i.e. gastrointestinal 
or respiratory disease” by for example gastrointestinal 
and respiratory disease, because there are many other endoscopic procedures.

DONE

Line 47: Please specify what is how low is the percentage of transient bacteremia in humans after endoscopy.

DONE

Lines 49-50: Please specify which patients are considered at risk and what procedures for antibiotic prophylaxis.

DONE

Line 51: Again if the author just mentioned one reference why this sentence says “Studies…” And what showed this particular study?

A reference was added. Their main findings and differences with our study are reported into the discussion section.

Line 55: Please delete “to add data”

DONE

Lines 55-56: The authors mention as a goal of this study: “assess whether or not antibiotic prophylaxis is required 
for such procedures in veterinary medicine” – but in fact the authors did not study that. They did not follow these patients clinically not even verified if the bacteremia was transient or persistent.

We stated into the discussion that no patients, even those who did not receive antibiotic therapy, developed symptoms attributable to infections or septicemia after the procedures. We can state that since all patients were clients commonly visited into the structure.

M&M

Line 63: The author’s mention that Dogs over six months…were potentially included. What does this mean exactly? Were they included or not?

Agreed, we have removed “potentially”

Line 69: The authors mention as a exclusion criteria the administration of antibiotics within the previous three weeks but then say that included dogs doing tylosin… This does not makes sense at all. These dogs should be obviously excluded from the study. This is an important bias for the results.

We disagree. Patients doing tylosin therapy have been included since tylosin was part, until recently, of the therapeutic and diagnostic trial of canine chronic enteropathies. Excluding those patients would have implied to exclude patients underwent gastro-duodeno-colonscopy. However, data about tylosin were statistically analysed as relative risk and no difference was found comparing patients with or without tylosin.

Lines 70-72: Why twenty minutes and not 30 min, 60, 120 minutes and/or 24h hours after? Is 20 minutes sufficient to bacteria translocate in enough numbers to be cultured? The choice of 20 minutes was based on two factors: 1) Human studies about bacteremia related to endoscopy report that in the majority of cases bacteremia is not detectable after 30 minutes (Shorvon et al., 1983 Gut 24: 1078-1093; Nelson, 2003 Gastrointest Endosc 57: 546-556); 2) The animals in this study were not experimental animals, so after the complete awakening from the anesthesia, the intravenous catheter was removed, and the patients were discharged from the hospital Is there any scientific data to support your choice? The answer to this question is above reported. How do the authors confirm if the bacteremia is transient or persistent? Into the limits’ paragraph just the first limit that we comment is the absence of a further blood sample. And therefore, how can the authors judge about the need (or not) of prophylactic antibiotics?? Seen the low incidence detected and the absence of symptoms attributable to infections or septicemia after the procedures, we strongly feel that the antibiotic should be advised only for therapeutic purpose and not prophylactic.

Lines 88-89: What type of cultures: aerobic, anaerobic or both?? Please clarify.

As known, the system uses a special broth medium which enables a wide range of aerobes, anaerobes and microaerophilic organisms to be cultured in a single bottle. We believe that it should not be added to the main text.

Results

Line 135: The authors mention that there were 8 patients in whom preprocedure results were positive and the postprocedure results were negative. How do the authors explain these tests results? Could they be false positives or false negatives? It seems to me that these results seriously question the culture methodology

Partially agreed. Other Authors (Li et al., 2012 Endosc Ultrasound 1: 90-95) performed the same study obtaining blood sample by using other methodologies, for example placing detained needles into the femoral artery which showed a 16% false positive blood cultures. A systematic review in human medicine (Dempsey et al., 2019 Am J Infect Control 47: 963-967) describes false positive ranges going from 0 to 11.7%. We have used a strong conservative approach. Blind subcultures were obtained from the negative blood cultures at the end of the 7th day by applying the same conditions used for the positive ones and none of those gave a positive result. For this reason, we can absolutely rule out the false negative results. On the other hand, as above referenced false positive results are surely possible showing a limit in specificity. Anyway, this limit has widely stressed into the discussion.

Line 157: Table 2: please insert the data regarding the number and percentage of cases for each variable

DONE

Discussion

Line 191-195: Taking in consideration the high number of these cases (the same number of positive cases) with contaminants as the author said, we may think that the collecting technic perhaps was not the most appropriate… and this may invalidate the results of this study. Any other explanation?

Already answered

Lines 220-221: Comparing this study with others, the authors mention that samples were taken from the intravenous catheter. Considering the results of the present study, it seems more feasible to avoid collecting the blood from a catheter used for other purposes, such as inject anesthesia medications. To me this increases the risk of getting a contaminated sample which again bias the results of this study.

Agreed, this limit has been discussed into the discussion section stating: “To eliminate additional painful and stressful manual manipulation, it was decided to use the intravenous catheter in order to minimize the impact on the animals, even if this represented another limitation.” Our study has been a field trial and not an experimental study, therefore patients had to be manipulated as less as possible.

Reviewer 3 Report

Reviewer comments for manuscript ID animals=999772 entitled ‘Incidence of bacteremia consequent to different endoscopic procedures in dogs: a preliminary study’

General comments

It is an interesting study that is quite relevant in context of the routine use of endoscopy in veterinary diagnostics. The patient load for undergoing such procedures is increasing and there is certainly a risk of bacterial contamination of endoscopic equipments that might lead to nosocomial infections. However, AMR in veterinary practice is a very critical issue and such studies reveal the unwarranted use of antibiotics. I congratulate the authors for conducting this study.

I have very few comments that I have specifically mentioned. I had this comment after reading the methodology section ‘I am of the opinion that the type of endoscopic procedure (bronchoscopy /gastroscopy/duodenoscopy) should have been included as a variable as breakage of protective mucosal barrier might depend on the part being studied’ but it stood clarified in the discussion. I have indicated that this should have been explained in the methodology section itself.

Discussion needs a more vigorous treatment through more literature search and indepth analysis of the results.

Specific comments

Summary

Lines 14-15: Please reframe ‘Also in Veterinary Medicine it is crucial to try to reduce the use of antimicrobials whenever they are not needed to slow down the process as part of the One Health concern’ as ‘There is a crucial need in Veterinary Medicine to reduce the use of antimicrobials to slow down the process and incidence of antimicrobial resistance as a one health concern’

Lines 19-20: Please delete ’Even if the results in the present study require additional studies’

Line 20: Please replace ‘first step’ with ‘first attempt’

Abstract

Line 23: Please replace ‘still under debate’ with ‘debatable’

Line 24: Please replace ‘drugs, avoiding administration’ with ‘antibiotics, avoiding their administration’

Line 25: Please replace ‘regarding’ with ‘highlighting’

Line 32: Please replace ‘as regarded’ with ‘with respect to’

Introduction

Line 49: Please replace ‘these provide for the’ with ‘that permit’

Line 52: Please replace ‘indications’ with ‘specific guidelines’

Line 55: Please reframe ‘to add data and to assess whether or not’ as ‘and to assess whether’

Materials and methods

Line 60: Please reword ‘were included’ as ‘were included in the study’

Line 63: Please reword ’potentially included’ as ‘were included in the study’

Lines 65-68: Exclusion of such patients should bias your study result. Such patients do require endoscopic procedures and decisions to use or not use antibiotics in such patients should be quite a relevant part of your study hypothesis. Please clarify.

Line 69: Why this exception was made ‘except for those in therapy with tylosin’?

Lines 72-74: The reason for taking the second blood sample, I think was to assess any breakage of the mucosal barrier by the endoscopic procedure carried out. Please clarify.

Line 81: Please replace ‘anesthesiologic’ with ‘anaesthetic’

Lines 151-53: This should have been described under the materials and methods section.

Discussion

Lines 168-69: Please delete ’Two blood samples for the detection of circulating bacteria were collected, before the procedure and twenty minutes after the end of the procedure’ as this is a repetition.

Lines 173-83: These are materials and methods which should be briefly described in the methodology section. Please restrict your discussion to your results.

Line 190: Please replace ‘components’ with ‘inhabitants’

Lines 207-08: Please provide a reference to this statement.

Line 211: Please reword ’assessed as 10.8% of incidence risk’ as ‘an incidence risk of 10.8%’

Line 211: Please reword ‘This agreed with the’ as ‘This is in agreement with the ‘

Lines 224-25: Please reword’ No patients, even those who did not receive antibiotic therapy, developed symptoms attributable to infections or septicemia after the procedures’ as ‘None of the patients developed symptoms attributable to infections or septicemia after the procedures’

Author Response

Reviewer comments for manuscript ID animals=999772 entitled ‘Incidence of bacteremia consequent to different endoscopic procedures in dogs: a preliminary study’

General comments

It is an interesting study that is quite relevant in context of the routine use of endoscopy in veterinary diagnostics. The patient load for undergoing such procedures is increasing and there is certainly a risk of bacterial contamination of endoscopic equipments that might lead to nosocomial infections. However, AMR in veterinary practice is a very critical issue and such studies reveal the unwarranted use of antibiotics. I congratulate the authors for conducting this study.

Thank you

I have very few comments that I have specifically mentioned. I had this comment after reading the methodology section ‘I am of the opinion that the type of endoscopic procedure (bronchoscopy /gastroscopy/duodenoscopy) should have been included as a variable as breakage of protective mucosal barrier might depend on the part being studied’ but it stood clarified in the discussion. I have indicated that this should have been explained in the methodology section itself.

We have indicated this into the Statistical analysis section (Lines:123-126) as rightly suggested.

Discussion needs a more vigorous treatment through more literature search and indepth analysis of the results.

Hopefully, we have adjusted the discussion as much as wished

Specific comments

Summary

Lines 14-15: Please reframe ‘Also in Veterinary Medicine it is crucial to try to reduce the use of antimicrobials whenever they are not needed to slow down the process as part of the One Health concern’ as ‘There is a crucial need in Veterinary Medicine to reduce the use of antimicrobials to slow down the process and incidence of antimicrobial resistance as a one health concern’

DONE

Lines 19-20: Please delete ’Even if the results in the present study require additional studies’

DONE

Line 20: Please replace ‘first step’ with ‘first attempt’

As also requested by the Reviewer #2, the sentence has been modified.

Abstract

Line 23: Please replace ‘still under debate’ with ‘debatable’

DONE

Line 24: Please replace ‘drugs, avoiding administration’ with ‘antibiotics, avoiding their administration’

DONE

Line 25: Please replace ‘regarding’ with ‘highlighting’

DONE

Line 32: Please replace ‘as regarded’ with ‘with respect to’

DONE

Introduction

Line 49: Please replace ‘these provide for the’ with ‘that permit’

DONE

Line 52: Please replace ‘indications’ with ‘specific guidelines’

DONE

Line 55: Please reframe ‘to add data and to assess whether or not’ as ‘and to assess whether’

DONE

Materials and methods

Line 60: Please reword ‘were included’ as ‘were included in the study’

DONE

Line 63: Please reword ’potentially included’ as ‘were included in the study’

DONE

Lines 65-68: Exclusion of such patients should bias your study result. Such patients do require endoscopic procedures and decisions to use or not use antibiotics in such patients should be quite a relevant part of your study hypothesis. Please clarify.

We think that including these peculiar categories could overestimate the incidence risk. In the future we will evaluate how much at-risk categories affect the chance to develop bacteremia after procedure compared to control group. Thank you for this input.

Line 69: Why this exception was made ‘except for those in therapy with tylosin’?

Patients doing tylosin therapy have been included since tylosin was part, until recently, of the therapeutic and diagnostic trial of canine chronic enteropathies. Excluding these patients would have implied to completely exclude patients that underwent gastro-duodeno-colonscopy. However, data about tylosin were statistically analysed as relative risk and no difference was found comparing patients with or without tylosin.

Lines 72-74: The reason for taking the second blood sample, I think was to assess any breakage of the mucosal barrier by the endoscopic procedure carried out. Please clarify

Agreed, we have specified

Line 81: Please replace ‘anesthesiologic’ with ‘anaesthetic’

DONE

Lines 151-53: This should have been described under the materials and methods section.

DONE

Discussion

Lines 168-69: Please delete ’Two blood samples for the detection of circulating bacteria were collected, before the procedure and twenty minutes after the end of the procedure’ as this is a repetition.

DONE

Lines 173-83: These are materials and methods which should be briefly described in the methodology section. Please restrict your discussion to your results.

DONE

Line 190: Please replace ‘components’ with ‘inhabitants’

DONE

Lines 207-08: Please provide a reference to this statement.

DONE

Line 211: Please reword ’assessed as 10.8% of incidence risk’ as ‘an incidence risk of 10.8%’

DONE

Line 211: Please reword ‘This agreed with the’ as ‘This is in agreement with the ‘

DONE

Lines 224-25: Please reword’ No patients, even those who did not receive antibiotic therapy, developed symptoms attributable to infections or septicemia after the procedures’ as ‘None of the patients developed symptoms attributable to infections or septicemia after the procedures’

DONE

Round 2

Reviewer 2 Report

The date of the Ethical Committee Letter is 24 November of 2020, and  the study included animals with procedures done since 2015.
From my point of view, the ethical evaluation of the study procedures should be done before the study and not after.